# Community health workers' barriers and facilitators to use a novel mHealth tool for motivational interviewing to improve adherence to care among youth living with HIV in rural Nepal

Rekha Khatri[1,2]*, Pragya Rimal[1,3], Maria L. Ekstrand[4], Sabitri Sapkota[1], Kripa Sigdel[1,5], Dikshya Sharma[1], Jene Shrestha[1], Srijana Shrestha[6,7], Bibhav Acharya[7,8]

1 Research and Innovation Function, Possible, Kathmandu, Nepal, 2 Sydney School of Health Sciences, Faculty of Medicine and Health, The University of Sydney, New South Wales, Australia, 3 Fielding School of Public Health, University of California, Los Angeles, Los Angeles, CA, United States of America, 4 Department of Medicine, Division of Prevention Science, University of California, San Francisco, San Francisco, California, United States of America, 5 Department of Psychology, Tribhuvan University, Kathmandu, Nepal, 6 Department of Psychology, Wheaton College, Norton, Massachusetts, United States of America, 7 Possible, New York, NY, United States of America, 8 Department of Psychiatry and Behavioral Sciences, University of California, San Francisco, San Francisco, California, United States of America

* rrekha.khatri@gmail.com

## Abstract

Adherence to treatment regimens is a common challenge in achieving HIV control, especially among youth. Motivational Interviewing (MI) is an evidence-based intervention to facilitate behavior change (such as adherence to treatment) by focusing on the client's priorities and motivations. Community Health Workers (CHWs), who are well situated to engage clients for care, can use MI but studies have shown that they often lose MI skills. While mHealth tools can support CHWs in delivering evidence-based counseling techniques such as MI, it is important to understand the barriers and facilitators in using such tools. Our parent study includes developing and testing a novel mHealth tool called, Community based mHealth Motivational Interviewing Tool for HIV-positive youth (COMMIT+). In this descriptive qualitative study, we share the results from semi-structured interviews with 12 CHWs who used COMMIT+ to engage youth living with HIV, and 7 of their Community Health Nurse supervisors. Our results demonstrate the barriers and facilitators experienced by CHWs in using a mHealth tool to deliver MI for youth living with HIV in rural Nepal, and highlight that supportive supervision and user-friendly features of the tool can mitigate many of the barriers.

**Data Availability Statement:** This study used qualitative data, and quotes from the participants

are included within the manuscript. The data contains personal identifiers and sensitive information of the participants and their clients, who are youth living with HIV, and these identifiers are interspersed in the whole transcript. Therefore, the data cannot be shared publicly in compliance with the regulations of the Ethical Review Board of the Nepal Health Research Council. For further information, please write to approval@nhrc.gov.np. For data access, please submit a reasonable request to the Research and Innovation department at Possible by emailing research@possiblehealth.org.

**Funding:** This study was funded by National Institute of Mental Health R34MH118049. BA and ME received the award. The funders had no role in study design, data collection and analysis, decision to publish, or preparation of the manuscript.

**Competing interests:** I have read the journal's policy and the authors of this manuscript have the following competing interests: Author Bibhav Acharya is currently serving as an Academic Editor at PLOS Global Public Health. Rest of the authors have declared that no competing interests exist.

## Introduction

Motivational Interviewing (MI) is an evidence-based intervention to facilitate healthy behaviors by focusing on the client's priorities and motivations [1–3]. It is an effective counseling approach that emphasizes partnership between clients and providers by creating an atmosphere of trust and acceptance [4]. MI has substantial evidence for effectively improving adherence to medications and regular clinic follow-up for multiple conditions [5] such as alcohol use, tobacco use, HIV, and diabetes in various care settings [2, 6–10].

MI has also been used by lay health workers, [11, 12] such as Community Health Workers (CHWs) who are well situated to engage clients for care improvement as they have a close understanding of, and a trusting relationship with, the communities they serve [13, 14]. Leveraging these strengths, CHWs have been trained to use MI in multiple countries [15, 16]. However, studies have shown that a common challenge for CHWs is that they lose MI skills few months after receiving training [17, 18]. This challenge is compounded by the fact that MI approaches can be counterintuitive (e.g., listening to the client's perspectives, even when the provider disagrees with them, and not giving advice unless the client explicitly asks for it) [19]. A potential opportunity to address this challenge of loss of skills is that CHWs are increasingly using digital tools to maintain other skills, to access and enter client data, and to connect with healthcare facilities [20–22].

In 2022, young people (15–24 years) accounted for 26% of all new HIV infections in the Asia and Pacific region [23]. Young people have poor retention in HIV health care services and experience high mortality [24]. Our global HIV efforts are falling behind on achieving the target of reducing the number of adolescent girls and young women aged 15–24 years who are newly infected with HIV globally every year [25]. Studies have pointed at external and internal barriers for youth to engage in care and adhere to them [26], yet there is lack of evidence-based strategies to improve care engagement among youth [27].

This paper focuses on CHWs' perspectives as part of a larger study which developed and tested a mobile health (mHealth) tool called Community based mHealth Motivational Interviewing Tool for HIV-positive youth (COMMIT+), using a 5-step human centered design, [28] to support CHWs in maintaining their MI skills. The tool had two components: it provided decision-support prompts to guide CHWs to use MI while engaging with youth living with HIV(YLWH) and it also allowed for recording consented conversations between CHWs and clients. CHWs later obtained feedback on these recordings from their supervisors, who are Community Health Nurses (CHNs) in our setting. The CHWs were trained and regularly supervised to deliver MI to youth, focusing on their adherence to medication and regular follow-up visits in health facilities. CHWs often lack this kind of ongoing supervision because they travel from one client's home to another without meeting their supervisors and COMMIT+ was designed to overcome to this challenge. Detailed description of the tool and the co-design process will be described in a forthcoming manuscript. Briefly, the tool was used by CHWs while counselling the clients. The tool guides the CHWs to engage with clients, prompting the CHWs to apply appropriate MI skills depending on the clients' stages of change. For example, if the client is in pre-contemplative phase, the tool prompts the CHWs to use "sustain talk" and ask open-ended questions to the clients. The tool has a recording feature, which allows the conversation between CHWs and the clients to be recorded, upon consent, for supervision purposes.

While digital health interventions such as mHealth tools are promising, it is important to assess their acceptability, feasibility, and additional factors that contribute to the success or pose challenges for optimal implementation and impact [29]. However, there is limited evidence regarding the barriers and facilitators for CHWs in delivering MI in community settings

using mHealth tools. This paper will illustrate the barriers and facilitators experienced by CHWs when using COMMIT+ to deliver MI for YLWH in rural Nepal.

## Methods

### Study site

This study was conducted among CHWs between 2018–2021 in Achham, a remote district in Nepal's far-western province, which has a high prevalence of HIV [30]. During the study period, *Possible*, a non-profit organization, managed Bayalpata hospital in Achham in collaboration with Nyaya Health Nepal and the Government of Nepal. The hospital system, which includes a CHW program, delivers primary care services to over 200,000 people in the region. CHWs in the study site are local women who have completed 10th grade, and are trained, paid, and supervised. CHWs conduct daily home visits and engage in awareness-raising and knowledge-sharing activities related to maternal and child care and a few non-communicable diseases [31, 32]. They use digital tools to support these activities and to collect data from the community members [33].

### Participants

All CHWs and CHNs had received a 35-hour MI skills training. CHNs received additional 20-hour MI supervision training. The key components of trainings were basics of client interaction, clients' stages of change, core techniques of MI, and importance of supervision. We conducted semi-structured 1:1 interview with 12 CHWs and 7 CHNs to understand their experiences, along with barriers and facilitators to using COMMIT+ for delivering MI to YLWH. We selected CHWs purposively from four different urban and rural municipalities of Achham to get a diverse sample of those with high and low number of visits (from a total CHW sample of 20). All CHWs had completed at least 3 visits per client. 4 CHWs had completed 4–6 visits and 3 CHWs had done 8–10 visits. The clients were recruited from 25th February, 2020 to 2nd April, 2021. All CHNs who were supervising the CHWs using COMMIT + were interviewed.

We conducted interviews at two time points during the study. The first round of interviews was conducted after 8 months of first participant recruitment. Initially, we interviewed 5 CHWs and 4 CHNs who were supervising the CHWs. Because of COVID related travel restrictions, some CHWs were unable to visit clients until several months later. To account for this, we did a second round of interviews to obtain insights for those CHWs as well. In this second round, we interviewed the 5 CHWs from the first round, and conducted additional interviews with 7 new CHWs and 3 new CHNs who were providing supervision.

### Tools and data collection

The study team members developed and finalized two semi-structured guides to conduct the interviews with CHWs and CHNs using similar approaches: beginning with open-ended questions followed by probes regarding their experiences of receiving training, understanding of MI, experiences in learning and using COMMIT+, receiving and providing supervision, and response of clients. We asked both CHWs and CHNs about general facilitators and barriers experienced by CHWs while engaging with the clients to deliver MI.

RK, MSW, who was not part of the MI training and supervision for CHWs and CHNs, conducted the interviews. Because of COVID-related travel restrictions, the interviews were conducted over the phone. After obtaining consent, the interviews were conducted on the preferred day and time of the participants. The interviews were recorded with the participants'

consent. Interview lasted between 20–55 minutes and were conducted in Nepali, the local language preferred by the participants. The interviewer RK is a female, native Nepali speaker, and did not have a supervisory role for the CHWs or the CHNs. She was employed as a Qualitative Research Manager by the research team. The participants were familiar with her as she had attended their meetings, interacted with them, and accompanied some of them for home visits to learn about the CHW program.

### Data analysis

We use the study staff and a professional service to transcribe and translate the interviews. The interviewer performed quality checks on all transcripts. The translators were provided with MI glossary [34]. The interviewer reviewed and compared initial Nepali and English transcripts, provided feedback to the translators, and reviewed the remaining English transcripts. After an initial review of the transcripts, we developed a code list, which was iteratively updated during the coding process. Coding was based on both a priori code book on potential barriers and facilitators, while allowing new codes to emerge, mainly in response to questions that asked about general reactions to the experiences of using the tool. We coded the data using Dedoose, a qualitative data management software. RK, the interviewer, coded all the data. After the coding was done, based on the similarity in the patterns of the codes, different themes and subthemes were derived from the data. The themes touched upon perception of MI, experiences of using MI, changes in MI skills, experiences of stigma, supervision experiences, usefulness of MI, and barriers faced by CHWs and facilitators for them to use COMMIT+. This paper focuses only on the barriers and facilitators for CHWs to use COMMIT+.

### Ethics approval

Ethical Approval from this study was obtained from Ethical Review Board of Nepal Health Research Council (Reg.no. 334/2018), University of California San Francisco (IRB # 18–25580, Reference # 224205) and Mount Sinai School of Medicine (IRB# 18–01400).

The qualitative data collection was done amidst the pandemic and therefore, we obtained verbal informed consent from all individual participants. The Ethical Review Board of Nepal provided approval for the same. To establish trust during virtual data collection, the interviewer made introductory calls with the participants and explained the purpose of the study and their role in it, following which interviews were conducted at the convenience of the participants. Before starting the interview, the interviewer read the consent script and obtained verbal informed consent from the participants. Since the interviews took place over the phone, the audio recording started only after the participant's consent, which was acknowledged by the interviewer and participant at the beginning of the interview.

## Results

A total of 19 participants (CHWs = 12; CHNs = 7) completed 28 interviews (9 participants in the first round and 19 in the second round). We present the identified themes as barriers and facilitators and include illustrative quotes.

### Barriers

Participants shared several barriers they faced when using COMMIT+. One important barrier was related to the pandemic as CHWs were unable to immediately start counseling after their training because of travel restrictions. CHWs noted experiencing many barriers during the initial stages, such as challenges in recording conversations with clients, and unlearning habits

from conventional counseling. Few of them also said that they did not feel as well-supported because even their supervisors lacked clarity on MI.

**Delay in delivering MI to clients.** CHWs and CHNs shared that they could not practice MI initially because of delays in client recruitment due to COVID-19 travel restrictions. Some CHNs reflected that even the CHWs who were very enthusiastic during the training lost confidence in their ability to deliver MI because of the delay. Some CHWs shared that when they were finally able to visit clients, they lacked the level of skills they had acquired during the training.

> *After our training, there was lockdown. We didn't follow-up [with clients] for some time and later, we followed up through the phone. I don't think we have been able to provide counseling according to MI [principles]. Maybe we have forgotten about it...Our trainers had given us training in such a good way but we weren't able to deliver it in the same way. CHW 08*

> *At that time, we received the training and we felt like we learned. I felt like, I can do this. Meanwhile, we didn't start seeing clients to use MI, so I forgot how to use this. Then, it became difficult. But [during the training], it had seemed so easy. MI feels so easy when we are learning but when we forget, then it is quite difficult and same old [MI-inconsistent] habit gets in the way. CHW 02*

**Unlearning habits from conventional counseling.** Prior to using MI, CHWs largely used advice-giving and educational approaches, so they knew what they were expected to say to the clients. However, because MI encourages exploration of the client's beliefs and priorities, and focuses on eliciting solutions from the client's, CHWs had difficulty unlearning the advice-giving approaches. They said that MI is an entirely new approach for them and they felt unsure about how to engage with clients. The CHWs shared that although they found MI to be useful, it was initially very difficult for them to deliver MI. They said that they didn't know what to say or how to start the conversation with the clients. Most of them shared that they were often confused and felt stuck in the middle of counseling. Few CHWs shared that they thought it was easier to use MI with clients who are eager to talk and share their experiences while it is harder with some clients, particularly children, who don't open up. With such clients, CHWs tended to revert back to advice-giving.

> *When receiving MI training, it was not so difficult for us. But in MI, there are so many ways to provide feedback, so many skills. I felt it was quite difficult knowing how to provide counseling to clients using all those skills. If you ask why, it is because we have never provided counseling according to MI. We cannot direct clients but allow them to find solutions by themselves. We cannot say, "You do this." And that is why, [we struggle] with what to say and how to use MI. CHW 01*

CHWs shared that they struggled to change the habits they had developed from their prior approach, which was telling clients what do to. They said despite being aware, sometimes they went back to doing what they used to do and started giving suggestions rather than asking clients for their preferences and priorities. The supervisors noted that the CHWs often struggled with this. They shared that there were long pauses in the client-CHW conversations as CHWs wanted to give advice, knew they should refrain, but were unsure about what to say next. Most CHNs shared that the CHWs were struggling to ask open-ended questions because they were used to asking closed-ended questions and giving advice.

**Hesitation in recording conversations.** CHWs and CHNs shared that there was initial hesitation in recording conversations from both the CHWs' and the clients' sides. Several CHWs shared concerns about recording the interaction because they had never been exposed to recording practices for supervision before. Because CHWs often felt unsure about how to respond to clients using MI skills, they were concerned that these silences and gaps would become evident in the audio recordings. Consequently, CHWs feared that the supervisors would negatively evaluate their performance after listening to the recordings.

*We had never recorded client conversations before. So, when I knew we had to record, I was a little taken aback. I was feeling worried. I was nervous about how to record, what to say. I started thinking my supervisor will evaluate me and what she will think of me. I became comfortable gradually but I used to feel worried in the beginning. CHW 01*

*They were feeling nervous that we will be hearing these conversations but later they become more comfortable. CHN 03*

CHWs eventually felt comfortable with the recordings as they experienced the benefits of receiving feedback from their supervisors without feeling judged. Both CHWs and their supervisors reported that the supervisors provided positive feedback when things went well. The supervisors also elicited responses from the CHWs themselves on what could be done better before directly sharing their constructive criticisms. The supervisors noted that this approach was different from how they were supervising earlier, when they had only focused on pointing out CHWs' mistakes.

CHWs shared that although all clients had consented to the study, including having the conversations recorded, CHWs initially sensed hesitation and fielded a lot of questions from the clients about the recording. Some CHWs also noted that some clients were initially self-conscious because of the recording.

*In my client's case, she asked me why we had to make a recording, what we will be doing with it. I reminded her that we won't do the recording without her permission. We explained everything . . . that it won't have their names and that [unauthorized] people will not hear it. I noticed that she doesn't open up as much when we started recording. When there is no recording, they will talk more openly but with recording they worry initially. CHW 09*

After the CHWs reminded the clients that the recordings will be confidential, the clients were reassured. CHWs described that clients' initial concerns were regarding what the recording could be used for and if the recordings were going to be made public. In one case, the client's father said that they have to be extra careful because if the news of his daughter's HIV infection spreads, it will be difficult for her to get married. The CHW and CHN who were assisting this client shared the importance of multiple visits to build trust with the family. After that, the client's father agreed to allow the audio recording.

**Limited opportunities to practice MI with YLWH.** Almost all CHWs and CHNs shared that they wished they had more clients to follow up to deliver MI as that would have given them more opportunity to practice. As this was a pilot study designed to give all CHWs some experience with using COMMIT+ and to determine acceptability and feasibility, the total number of clients per CHW was low. Consequently, CHWs mentioned that once the clients are in maintenance stage (i.e., clients are adherent to treatment), CHWs did not have opportunities to practice using MI skills with YLWH who were in other stages of change.

**Lack of MI skills among the supervisors.** A few CHWs shared that they felt that their supervisors were not as skillful in using MI and therefore weren't able to provide adequate support. Those CHWs felt that if their supervisors were more skilled, it would have been easier for them to use MI. A CHW shared:

*I feel that my supervisor hesitates when I ask her something about MI. . . . . . . I had told her I got stuck at this point and what I should be saying there. I said if we could do role-play where she could counsel me using MI. But she hesitated. CHW 05*

In the first round of interviews, a few supervisors shared they did not feel entirely confident about supervising CHWs. In the second round, all supervisors noted that they felt more comfortable in supervising CHWs based on the supervision guidelines and via the support of the study psychologist, with whom they regularly discussed their challenges.

## Facilitators

Participants listed several factors that helped CHWs gain comfort and confidence in delivering MI using COMMIT+, including training and practice, supervision support, features of COMMIT+, and peer learning among CHWs, as discussed in detail below.

**Training and practice.** CHWs and CHNs shared that the trainings were helpful for them to learn what to do and how to use COMMIT+. Some CHNs shared that before visiting clients, they often referred to and reviewed the training materials and the tool together with the CHWs. Training materials included slide decks, notes, and lecture videos [35, 36].

*We practiced again [using COMMIT+]. We discussed where we need to focus to ensure the clients do not return to contemplative and pre-contemplative phase from the maintenance phase. We also looked over the slides and discussed where the gaps were. CHN 03*

*We spoke to our supervisor in between [seeing clients] and asked them about how to do [MI]. We also received [refresher] trainings in between. . . . The more we practiced, the easier it became. CHW 01*

Some CHWs shared that they initially felt confident that they could deliver MI during the training. However, it was difficult to implement it in the beginning but the more they did it, they started becoming comfortable.

*In the beginning, we received the training but until we started practicing, everything was difficult. We couldn't counsel smoothly, but as we practiced more, we learned what this is about. It felt easier as we kept doing it. CHW 09*

**Supervision support.** Most CHWs reported supervisors' support was crucial for them to use MI. CHWs and CHNs shared that they both listened to the CHW-client audio recordings made by CHWs with the clients. CHNs then noted down their observations and discussed specific MI skills that the CHWs should be utilizing during counseling. CHNs reported using MI principles, such as eliciting responses from the participants themselves, during the supervision process by asking CHWs about their counseling experience, what went well, and potential areas for improvement. CHWs noted that after listening to the recording, their supervisors appreciated what they had done well, and also gave them specific feedback, such as pointing out when they were using closed-ended questions and encouraging them to ask more open-

ended questions, or where they could use another MI skill such as reflective listening. The CHWs would also share their difficulties with the CHNs, such as feeling stuck in the middle of counseling or being confused about how to use different MI skills, and the CHNs provided guidance to the CHWs on how to proceed. Some CHWs shared that in rare cases CHNs accompanied some CHWs' first visit to observe their counseling skills and provide feedback.

A few CHNs highlighted the importance of role-plays as part of supervision. A CHN shared that she used to spend an hour doing MI-focused role-plays with the CHWs before visiting the client.

*I requested CHWs to call me before follow-up with the client. We did role-plays where I became the client and CHW did counseling. CHN 04*

**Features of COMMIT+.** All CHWs and CHNs shared that COMMIT+ helped the CHWs when delivering MI. CHWs said that the decision-support prompts in the tool helped them if they forgot anything and provided them direction on moving the counseling session forward. CHNs also shared that CHWs found the tool useful because the prompts helped the CHWs about what to say or ask and which MI skills to use, without which the CHWs would have felt stuck.

*It is easier to look at the tool and do counseling rather than doing without it. If there were no tool, then it would have been confusing to know what to say when. The tool helps us to know what to say. It is useful. CHW 05*

*When we come back from the training, we have so many other things to take care of and we tend to forget. Because of the tool, we know when to speak, when the clients should speak, when to listen to the clients. There isn't anything that I wished wasn't there in the tool. CHW 06*

Participants liked the ability of the tool to record the conversations between the CHW and the clients. These recordings were shared with the CHNs, based on which CHNs provided feedback to the CHWs.

*We listen to the recording and then pause it to discuss what types of MI techniques were used [by the CHW]. The CHW tells me "I used this certain type". And then, we see if there are other places where MI techniques could be used but the CHW didn't do so, then I say, "we could do like this here." We both discuss what we could say to the clients. We listen, pause the recording, [and reflect] in this way. CHN 01*

*After listening to the recording, my supervisor asked me what went well and where I need to improve, I told her [what I thought] and she added few other things and asked what I thought of them. She pointed out the areas I could improve. She supports me like that and this is why it has been easier for me. CHW 02*

CHNs also shared that the tool supported them to provide feedback to the CHWs.

*During the training, I felt like this was too complicated and was wondering how to do it, but when I went through the tool couple of times, I found that we can easily give feedback. CHN 03*

**Peer learning.** CHWs discussed their experience of using MI with one another and noted that doing so was helpful for them. They shared the importance of peer discussion in sharing the challenges they faced and asking for suggestions to use COMMIT+ and/or MI skills. CHWs shared that they contacted peers to ask how they responded to clients in a particular situation, how they recorded and also to help clarify any confusions about using COMMIT+. A CHW shared that she received advice on how to effectively engage with clients from a peer CHW who had been utilizing MI techniques for a longer duration.

*We talk about the questions that we can ask the clients using MI . . . how to do it . . . share that it is difficult to do so and how we aren't perfect. We also discuss the difficulty in practicing affirmation and acceptance. . . we also talk about how we had been doing counseling in a way [that is different from MI]. CHW 10*

Few CHWs also shared their experiences of role-playing with peers to practice using COMMIT+ for MI.

*We met to practice counseling. We did role-play and took turns becoming the client and the CHW. We did the audio recording as well. . . ..it was helpful. She knows more about MI than I do and I could learn from her. CHW 03*

## Discussion

The barriers and facilitators that CHWs and CHNs reported in our study demonstrated multiple challenges and opportunities of relevance in other similar settings. COMMIT+ introduced several new tasks that initially presented as barriers for CHWs but it is encouraging to note that these barriers were reduced over time.

Feedback based on direct observation is a critical component of psychotherapy training [37, 38]. However, because CHWs are not based at a facility where they could potentially be observed when interacting with a client, our team had developed COMMIT+ to facilitate audio recordings. However, this approach had never been used at the research site, either in facilities or at client's homes, and this appears to have resulted in an initial barrier as both CHWs and clients were cautious and hesitant about recordings, as has been found in other settings as well [39–41]. However, as trust developed—both between clients and CHWs and between CHWs and CHNs, the audio recording enabled by the tool was considered as a facilitator as it supported the supervision process. This highlights the importance of trust with clients and psychological safety [42, 43] in the clinical team to make a novel tool become acceptable and useful. When working with populations that face stigma and discrimination, such as YLWH, confidentiality concerns are expected, even after they have consented for participation. It is likely that CHWs were able to build rapport with the clients and their families given their strong presence in the community and being seen as trusted members of the healthcare team [44]. Our experiences have also shown the need to engage with the family members of YLWH, especially in the South Asian context, where parents are the primary caregivers and decision makers and their voices become important for young people to participate in studies [45, 46].

Based on our direct observations during training and support for CHWs prior to the study, [19, 34] and the broader literature on CHW engagement in the communities, [18, 47, 48] it is clear that CHWs are often involved in advice-giving as a way to encourage healthy behaviors. After the 35-hour MI training, although they had demonstrated using MI principles (such as

following the patient's lead rather than educating them), they continued to revert back to advice-giving when they struggled to respond to clients. This has been seen in other studies where CHWs are trained in using MI [18]. This barrier was reduced through time and the process of improvement appears to have been facilitated by the use of COMMIT+, including reviewing the audio-recordings by the supervisors.

Despite the overwhelming evidence of MI's effectiveness in behavior change, it is still not widely available in low-resource settings. This presented a challenge in our study as the supervisors were also learning MI for the first time [49]. Although using dedicated clinical psychologists (who have had prior MI training) would have provided a robust supervisory system, this would not have been scalable given the dearth of specialists in LMICs. As such, one barrier to scale-up COMMIT+ is that even the supervisors were sometimes not comfortable using MI. However, the qualitative data show that this also improved through time but it highlights the need for more support to supervisors [50]. This finding also implies the necessity of offering MI training with adequate support to other health workers in supervisory roles to address the shortage of specialists in LMICs.

The primary facilitator for CHWs was in experiencing the benefits of maintaining their skills via supportive supervision from CHNs. Supportive supervision improves CHWs' motivation [51, 52] and in our study, it appears to have provided a lot of positive reinforcement for CHWs (and CHNs) to continue to use COMMIT+ as they could directly see the benefits. This is consistent with the literature on psychotherapy training that benefits from deliberate practice and clinical supervision [38, 40, 41]. One finding that we had not anticipated is that learning MI helped the supervisors to be more supportive and less judgmental, demonstrating that teaching MI for use with clients can create a positive spillover effect where the healthcare workers use MI principles in their interactions with each other.

As human-centered design [28] was used and both YLWHs and CHWs were part of the design process, the tool included features that were considered to be helpful for the CHW end users. The prompts, specific text, and lack of distracting information likely helped the CHWs use the tool to address challenges they anticipated while delivering MI. This finding is consistent with the broader best practice of involving end users/final beneficiaries in the process of developing digital tools [53, 54].

CHWs highlighted the importance of accessing materials from their MI training as a facilitator. Materials that summarize the training content, which can be easily referred to after the training, can support the implementers. Although the tool only had high-yield items that CHWs can view while interacting with clients, this finding may indicate the importance of adding the training materials in a different part of the tool that can be used as a reference. It is possible that using this strategy may have increased ease of use of motivational interviewing, and this is something that can be tested in future studies.

Our findings highlight the importance of peers in maintaining newly acquired skills that are often very different from conventional approaches. Strategies for peer-based supervision have been shown to be effective in other behavioral interventions as well in LMICs [55, 56]. Although we had not emphasized this aspect in our study, the CHWs informally access peer-based learning. As such, our study may have been unable to uncover the impact of such peer-based support. This can be formally incorporated into future studies by building in structures to foster and study peer supervision.

For future research studies, especially among sensitive research populations, it seems important to take cultural contexts into consideration and employ strategies that support community workers to adequately build rapport and trust with research participants, while rolling out digital tools. Certain features of the mHealth tools—the audio recording feature in our case—can be concerning for end users leading to initial resistance but our data has shown that

supportive supervision with a non-judgmental learning environment can be effective at overcoming the barriers experienced by end users. However, future studies can assess implementation strategies that will increase the likelihood of initial uptake of novel digital health features.

## Conclusion

In summary, we identified several initial barriers to the implementation of the digital intervention COMMIT+ but were able to overcome them through practice and supervision. This study provides preliminary evidence that mHealth interventions that are designed by incorporating the perspectives of healthcare providers can assist them in maintaining skills to use a new behavioral technique, especially for CHWs, who often do not receive adequate support and supervision despite having an immense potential to implement evidence-based interventions for vulnerable populations in many global settings with limited resources.

## Acknowledgments

The authors would like to acknowledge the support provided by Nyaya Health Nepal, a not for profit organization in Nepal, in this study. We also thank all the participants of the study.

## Author Contributions

**Conceptualization:** Maria L. Ekstrand, Bibhav Acharya.

**Formal analysis:** Rekha Khatri.

**Funding acquisition:** Maria L. Ekstrand, Bibhav Acharya.

**Investigation:** Rekha Khatri.

**Methodology:** Rekha Khatri, Pragya Rimal.

**Project administration:** Pragya Rimal, Sabitri Sapkota.

**Resources:** Sabitri Sapkota, Bibhav Acharya.

**Supervision:** Sabitri Sapkota, Bibhav Acharya.

**Writing – original draft:** Rekha Khatri.

**Writing – review & editing:** Pragya Rimal, Maria L. Ekstrand, Sabitri Sapkota, Kripa Sigdel, Dikshya Sharma, Jene Shrestha, Srijana Shrestha, Bibhav Acharya.

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
