## [Decision Letter · Decision Letter 0]

24 Nov 2023

PGPH-D-23-01981

Community Health Workers’ barriers and facilitators to use a novel mHealth tool for motivational interviewing to improve adherence to care among youth living with HIV in rural Nepal

Dear Dr. Khatri,

Thank you for submitting your manuscript to PLOS Global Public Health. After careful consideration, we feel that it has merit but does not fully meet PLOS Global Public Health’s publication criteria as it currently stands. Therefore, we invite you to submit a revised version of the manuscript that addresses the points raised during the review process.

Kindly pay attention to addressing the comments provided by reviewer 1.

We look forward to receiving your revised manuscript.

Kind regards,

Ferdinand Mukumbang, PhD

Academic Editor

Journal Requirements:

1. In the ethics statement in the Methods, you have specified that verbal consent was obtained. Please provide additional details regarding how this consent was documented and witnessed, and state whether this was approved by the IRB

2. Please send a completed 'Competing Interests' statement, including any COIs declared by your co-authors. If you have no competing interests to declare, please state "The authors have declared that no competing interests exist". Otherwise please declare all competing interests beginning with the statement "I have read the journal's policy and the authors of this manuscript have the following competing interests:"

Additional Editor Comments (if provided):

Reviewers' comments:

Reviewer's Responses to Questions

**Comments to the Author**

1. Does this manuscript meet PLOS Global Public Health’s publication criteria? Is the manuscript technically sound, and do the data support the conclusions? The manuscript must describe methodologically and ethically rigorous research with conclusions that are appropriately drawn based on the data presented.

Reviewer #1: Yes

Reviewer #2: Yes

2. Has the statistical analysis been performed appropriately and rigorously?

Reviewer #1: N/A

Reviewer #2: N/A

3. Have the authors made all data underlying the findings in their manuscript fully available (please refer to the Data Availability Statement at the start of the manuscript PDF file)?

Reviewer #1: Yes

Reviewer #2: Yes

4. Is the manuscript presented in an intelligible fashion and written in standard English?

Reviewer #1: Yes

Reviewer #2: Yes

5. Review Comments to the Author

Reviewer #1: Overall, this is an interesting and useful paper, but it can be further improved by adding some methodological and intervention-related details and justifications regarding the analysis approach. I think the approach seemingly taken makes sense for this kind of intervention-focused study, but the reader currently has to fill in some blanks about the qualitative methods.

The article is well-written, provides a useful background, and most parts of the process are well explained and justified. You could also mention this predominantly being a pragmatic/directed/descriptive qualitative study. You have not really mentioned in the abstract either what kind of study this is – “results of semi-structured interviews” does not tell the reader very much yet. Did you use some kind of qualitative reporting guidelines, e.g., SRQR? That might help to ensure all crucial details are included (many of which are, such as the role/relationship of the interviewer etc.).

Some more details on the analysis approach in particular would be helpful. Did you use any specific analysis methods that you could cite (e.g., what kind of thematic analysis?), or if not, could you explain more about the approach you developed yourselves? E.g., what guided coding and theme development – was it very data-driven/deductive or mainly informed by the “search” for barriers and facilitators? Rather state this explicitly than having the reader guessing based on your analytic output. Remember, if your interview guides mainly covered barriers and facilitators, then it will not sound coherent to describe the analysis as open-ended and inductive or similar while reporting on mainly barriers and facilitators. Similarly, what significance does it have that you “single-coded” – does this mean one person did all the coding once? Clarify and justify.

When reading the discussion, I wonder if providing a bit more information about the COMMIT+ tool itself early on in the article (e.g., how was it used by CHWs/CHNs in practice, example prompts and a bit more description about what the CHWs and CHNs wanted it to include in the HCD/co-design phase, were they keen on the recording then for example?) would also help contextualise the findings and discussion? I assume this has been published in detail elsewhere, but some kind of text box/figure/similar or just a few more sentences in the main text would be beneficial here too.

Lastly, I am pleased to see an example of successfully (even if not without challenges) introducing new responsibilities/approaches to CHWs’ work, as this is often done without consultation and with the expectation that volunteer or low-paid CHWs will just keep taking on more and more tasks (which understandably tends to fail).

Reviewer #2: A very timely and pertinent paper targeting the most vulnerable population i.e the youth who are on HIV care. The paper is well written but there is need to follow the COREQ reporting guidelines for qualitative work especially on domains 1 (research team and reflexivity) and domain 3 (analysis and findings).

Also clarify if the 5 CHNs interviewed twice if there was a difference and if the results for both interviews were used for this manuscript or not.

6. PLOS authors have the option to publish the peer review history of their article (what does this mean?). If published, this will include your full peer review and any attached files.

**Do you want your identity to be public for this peer review?** For information about this choice, including consent withdrawal, please see our Privacy Policy.

Reviewer #1: No

Reviewer #2: No

---

## [Decision Letter · Decision Letter 1]

5 Mar 2024

PGPH-D-23-01981R1

Community Health Workers’ barriers and facilitators to use a novel mHealth tool for motivational interviewing to improve adherence to care among youth living with HIV in rural Nepal

Dear Dr. Rekha Khatri,

Thank you for submitting your manuscript to PLOS Global Public Health. After careful consideration, we feel that it has merit but does not fully meet PLOS Global Public Health’s publication criteria as it currently stands. Therefore, we invite you to submit a revised version of the manuscript that addresses the points raised during the review process.

One of the reviewers have requested more nuanced discussion on the implications of the study findings and the limitations of the study in light of how they impacted the study findings. Kind addresses these concerns.

We look forward to receiving your revised manuscript.

Kind regards,

Ferdinand Mukumbang, PhD

Academic Editor

Journal Requirements:

2. Please send a completed 'Competing Interests' statement, including any COIs declared by your co-authors. If you have no competing interests to declare, please state "The authors have declared that no competing interests exist". Otherwise please declare all competing interests beginning with the statement "I have read the journal's policy and the authors of this manuscript have the following competing interests:"

Additional Editor Comments (if provided):

Reviewers' comments:

Reviewer's Responses to Questions

**Comments to the Author**

1. If the authors have adequately addressed your comments raised in a previous round of review and you feel that this manuscript is now acceptable for publication, you may indicate that here to bypass the “Comments to the Author” section, enter your conflict of interest statement in the “Confidential to Editor” section, and submit your "Accept" recommendation.

Reviewer #1: All comments have been addressed

Reviewer #2: All comments have been addressed

2. Does this manuscript meet PLOS Global Public Health’s publication criteria? Is the manuscript technically sound, and do the data support the conclusions? The manuscript must describe methodologically and ethically rigorous research with conclusions that are appropriately drawn based on the data presented.

Reviewer #1: Yes

Reviewer #2: Yes

3. Has the statistical analysis been performed appropriately and rigorously?

Reviewer #1: N/A

Reviewer #2: N/A

4. Have the authors made all data underlying the findings in their manuscript fully available (please refer to the Data Availability Statement at the start of the manuscript PDF file)?

Reviewer #1: Yes

Reviewer #2: No

5. Is the manuscript presented in an intelligible fashion and written in standard English?

Reviewer #1: Yes

Reviewer #2: Yes

6. Review Comments to the Author

Reviewer #1: (No Response)

Reviewer #2: I appreciate the authors' thorough response to the initial round of feedback. They have successfully addressed several concerns, including the improved description of the methodology.The discussion section is more robust, but I encourage the authors to delve deeper into the implications of their findings for future research and policy. Consider discussing potential strategies informed by the study.While the limitations are adequately addressed, providing a more nuanced discussion on how these limitations may have influenced the study's outcomes would enhance the manuscript's transparency.I want to acknowledge the authors' diligence in revising the manuscript based on the initial feedback, and I believe that these additional suggestions will contribute to the manuscript's overall strength. Thank you for considering my second review, and I look forward to witnessing the continued development of this valuable contribution to the field.

7. PLOS authors have the option to publish the peer review history of their article (what does this mean?). If published, this will include your full peer review and any attached files.

**Do you want your identity to be public for this peer review?** For information about this choice, including consent withdrawal, please see our Privacy Policy.

Reviewer #1: No

Reviewer #2: No

---

## [Decision Letter · Decision Letter 2]

17 Jun 2024

Community Health Workers’ barriers and facilitators to use a novel mHealth tool for motivational interviewing to improve adherence to care among youth living with HIV in rural Nepal

PGPH-D-23-01981R2

Dear Ms. Khatri,

We are pleased to inform you that your manuscript 'Community Health Workers’ barriers and facilitators to use a novel mHealth tool for motivational interviewing to improve adherence to care among youth living with HIV in rural Nepal' has been provisionally accepted for publication in PLOS Global Public Health.

Best regards,

Julia Robinson

Executive Editor

Reviewer Comments (if any, and for reference):

Reviewer's Responses to Questions

**Comments to the Author**

1. If the authors have adequately addressed your comments raised in a previous round of review and you feel that this manuscript is now acceptable for publication, you may indicate that here to bypass the “Comments to the Author” section, enter your conflict of interest statement in the “Confidential to Editor” section, and submit your "Accept" recommendation.

Reviewer #1: All comments have been addressed

Reviewer #2: All comments have been addressed

2. Does this manuscript meet PLOS Global Public Health’s publication criteria? Is the manuscript technically sound, and do the data support the conclusions? The manuscript must describe methodologically and ethically rigorous research with conclusions that are appropriately drawn based on the data presented.

Reviewer #1: Yes

Reviewer #2: Yes

3. Has the statistical analysis been performed appropriately and rigorously?

Reviewer #1: N/A

Reviewer #2: N/A

4. Have the authors made all data underlying the findings in their manuscript fully available (please refer to the Data Availability Statement at the start of the manuscript PDF file)?

Reviewer #1: Yes

Reviewer #2: Yes

5. Is the manuscript presented in an intelligible fashion and written in standard English?

Reviewer #1: Yes

Reviewer #2: Yes

6. Review Comments to the Author

Reviewer #1: (No Response)

Reviewer #2: A very timely manuscript and especially in longterm adherence to medication.

7. PLOS authors have the option to publish the peer review history of their article (what does this mean?). If published, this will include your full peer review and any attached files.

**Do you want your identity to be public for this peer review?** For information about this choice, including consent withdrawal, please see our Privacy Policy.

Reviewer #1: No

Reviewer #2: No
